# IHCScoreGAN: An unsupervised generative adversarial network for end-to-end Ki67 scoring for clinical breast cancer diagnosis

**Carl P. Molnar**[1]                                                                  MOLNAR.CARL@MAYO.EDU

**Thomas E. Tavolara**[1]                                                      TAVOLARA.THOMAS@MAYO.EDU

**Christopher A. Garcia**[1]                                               GARCIA.CHRISTOPHER@MAYO.EDU

**David S. McClintock**[1]                                                  MCCLINTOCK.DAVID@MAYO.EDU

**Mark D. Zarella**[1]                                                             ZARELLA.MARK@MAYO.EDU

**Wenchao Han**[1]                                                                HAN.WENCHAO@MAYO.EDU

[1] *Division of Computational Pathology and AI, Mayo Clinic, Rochester, USA*

**Editors:** Accepted for publication at MIDL 2024

## Abstract

Ki67 is a biomarker whose activity is routinely measured and scored by pathologists through immunohistochemistry (IHC) staining, which informs clinicians of patient prognosis and guides treatment. Currently, most clinical laboratories rely on a tedious, inconsistent manual scoring process to quantify the percentage of Ki67-positive cells. While many works have shown promise for Ki67 quantification using computational approaches, the current state-of-the-art methods have limited real-world feasibility: they either require large datasets of meticulous cell-level ground truth labels to train, or they provide pre-trained weights that may not generalize well to in-house data. To overcome these challenges, we propose IHCScoreGAN, the first unsupervised deep learning framework for end-to-end Ki67 scoring without the need for any ground truth labels. IHCScoreGAN only requires IHC image samples and unpaired synthetic data, yet it learns to generate colored cell segmentation masks while simultaneously predicting cell center point and biomarker expressions for Ki67 scoring, made possible through our novel dual-branch generator structure. We validated our framework on a large cohort of 2,136 clinically signed-out cases, yielding an accuracy of 0.97 and an F1-score of 0.95 and demonstrating substantially better performance than a pre-trained state-of-the-art supervised model. By removing ground truth requirements, our unsupervised technique constitutes an important step towards easily-trained Ki67 scoring solutions which can train on out-of-domain data in an unsupervised manner. Our code and model weights are available at `https://github.com/WenchaoHan0718/IHCScoreGAN`.

**Keywords:** Generative Adversarial Networks, Unsupervised Learning, Computational Pathology, Ki67 Scoring, Breast Cancer

## 1. Introduction

Ki67 is a protein biomarker whose percentage of Ki67-positive tumor cells has shown to be effective for indicating tumor proliferation in breast cancer cases (Davey et al., 2021). In a clinical setting, the Ki67 score is reported routinely for breast cancer IHC tissue samples to aid in diagnosis and guide treatment decisions (Faneyte et al., 2003; Chang et al., 2000; Petit et al., 2004). Currently, in most centers, Ki67 scoring is manually performed by pathologists, which is time-consuming and subject to inter-observer variability (Reisenbichler et al., 2020). Therefore, automatic Ki67 scoring for breast tissue samples are extremely desirable.

Unfortunately, automatic stain scoring is challenging due to the prohibitive data demands for training or fine-tuning supervised deep learning-based IHC scoring models. These models require pixel-level annotations of hundreds of thousands of cells, painstakingly labeled by highly trained technicians (Abousamra et al., 2020; Fassler et al., 2020; Graham et al., 2019; Wen et al., 2023; Van Eycke et al., 2018; Priego-Torres et al., 2022; Zhang et al., 2020). DeepLIIF (Ghahremani et al., 2022) is a supervised framework which instead trains using multiple co-registered multiplex immunofluorescence (mpIF) images as its ground truth to eliminate cell annotation errors. However, mpIF assays are expensive and not widely available. On the other hand, pre-trained supervised models are rarely publicly available and not always applicable to in-house stain data: tissue stains are collected from a particular area of the body with distinct tissue features, using a unique arrangement of best practices, tissue scanners, chemical mixtures, and tissue qualities, resulting in a unique in-house data distribution (Wagner et al., 2024; Therrien and Doyle, 2018).

Recently, several studies have emerged in an attempt to completely eliminate the need for ground truth labels. To this end, unsupervised deep learning methods typically leverage CycleGAN (Zhu et al., 2017), a well-known framework for unpaired and unsupervised domain transfer. One such example is automatic stain transfer, where the goal is to transfer IHC stain images into hematoxylin-eosin (H&E) stain images in an unsupervised manner (Liu et al., 2021; Trullo et al., 2022; Lin et al., 2023). However, performing a stain transfer from IHC to H&E loses the biomarker expression information associated with the IHC stain, making IHC scoring impossible. Unsupervised nuclear semantic segmentation (Le Bescond et al., 2022) and instance segmentation (Wang et al., 2023) were recently shown to be possible by leveraging binary cell segmentation masks created from H&E images; however, both of these approaches lack the ability to retain vital IHC biomarker expression information. In the field, there remains an unmet need for an end-to-end IHC scoring method which can avoid the need for ground truth labeling.

In this paper, we propose IHCScoreGAN, the first unsupervised deep learning framework to provide end-to-end scoring for IHC sample images. We leverage a domain transfer strategy to formulate a novel learning task, along with a novel model architecture which facilitates this task with a dual-branch generator. Our learning task seeks to transfer Ki67 images into synthetic cell segmentation masks extracted from an unpaired public dataset of H&E tissue images, instilling cell center points and synthetic cell colors. Significantly, our model automatically learns the correspondence between the Ki67-positive and -negative cells and the synthetic cell colors while also generating center point predictions, which together provides a predicted Ki67 score. We validated our framework on a large cohort of clinical cases sourced from Mayo Clinic, demonstrating strong performance on 2,126 clinically signed-out breast cancer cases collected over the span of 11 years, and further validated on an external dataset for cell counting, yielding competitive performance against supervised methods. Our proposed contributions are as follows:

- We propose a novel domain transfer strategy for unsupervised, unpaired end-to-end IHC scoring, where the learning task is constructed from synthetic colored cell segmentation masks and center points easily extracted from public H&E images.
- We propose a novel model architecture to accommodate our learning task, which can generate colored cell segmentation masks as a proxy task while simultaneously generating supplementary information for IHC scoring during inference.

## 2. Methodology

Our proposed framework, IHCScoreGAN, is trained by leveraging a novel learning task constructed from unpaired, publicly-sourced H&E data. It does not require any preprocessing or postprocessing of IHC stain images (Figure 1a). To facilitate our learning task, we build a novel model architecture which achieves end-to-end Ki67 scoring by splitting our learning task into two parts: 1) its primary goal is to generate a colored cell segmentation mask, which serves as a proxy task for 2) predicting cell center points and biomarker expressions (Figure 1b). From this information we can extract a Ki67 score, which is aggregated per slide and evaluated against clinically-derived scores (Figure 1c).

### 2.1. Dataset

#### 2.1.1. Internal IHC Dataset

Our internal dataset consists of 2,126 Ki67 digitized slides, each corresponding to a distinct breast cancer case. All cases in this study are sourced from Mayo Clinic and collected from 2012 through 2023, primarily by 6 pathologists. The Ki67 slides were scanned at $20\times$ (0.5µm/pixel) magnification by Aperio® AT scanners. Clinical diagnoses were performed on selected invasive tumor regions. Typically, one pathologist is involved in the clinical diagnosis for each case. We divided each Ki67 tissue slide into tiles of size $256 \times 256$ pixels within the selected regions, resulting in $N_{\mathrm{IHC}} = 678,134$ total tiles. We formally define our internal IHC dataset as $S_{\mathrm{IHC}} = \{x_i\}_{i=1}^{N_{\mathrm{IHC}}}$, where $x \in \mathbb{R}^{256\times256\times3}$ is a Ki67 stain tile.

#### 2.1.2. Target Dataset

We generated our target dataset using unpaired H&E-stained tissue slides sourced from The Cancer Genome Atlas (TCGA) database (Weinstein et al., 2013), a public clinical data repository. We randomly sampled H&E tissue slides of $20\times$ magnification from the Breast Invasive Carcinoma project (TCGA-BRCA) and manually selected slides to represent a variety of cell sizes, structures, and shapes (i.e., to discourage the model from using these features to discriminate real vs. fake data), resulting in 23 slides. We then divided each slide into tiles of size $256\times256$ and sampled $N_{\mathrm{MASK}} = 10,000$ total tiles. Next, we passed the H&E tiles through a publicly available HoVerNet (Graham et al., 2019) model, pre-trained on the CPM-17 dataset (Vu et al., 2019) to predict cell instance contours.

    We generated synthetic colored cell segmentation masks for each H&E tile by assembling the predicted cell instance contours into a cell segmentation mask, drawing a ratio from a uniform distribution $\mathcal{U}_{[0,1]}$, and then randomly picking cells to color green based on the ratio and coloring the rest red. We also draw a value from a uniform distribution $\mathcal{U}_{[0,1]}$ for each cell and multiply the cell's pixel intensities by this value, in order to emulate different IHC biomarker expression intensities (see "Real Y" in Figure A.1). We then generated distinct binary masks for our two synthetic cell colors, which detach cell expression prediction from the intensity values. Finally, we generated cell center point distance maps, instilling 2-norm distances between each pixel in each cell and its corresponding predicted center point.

    We formally define our external target dataset as $S_{\mathrm{MASK}} = \{(y_i, k_i)\}_{i=1}^{N_{\mathrm{MASK}}}$, where $y \in \mathbb{R}^{256\times256\times3}$ is a synthetic colored segmentation mask tile and $k \in \mathbb{R}^{256\times256\times3}$ is its matching cell center point distance map and two binary cell expression masks.

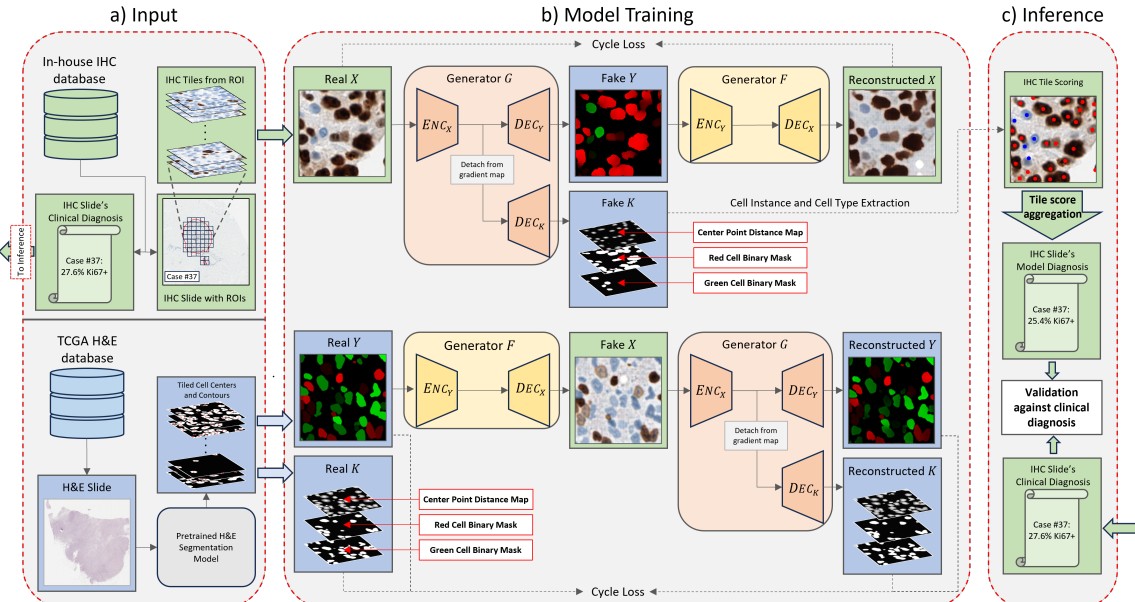

Figure 1: Overview of our proposed framework. a) Ki67 images, which have associated diagnostic data, are tiled (top). Unpaired H&E slides are passed through a model which predicts cell centers and contours; from this, we produce synthetic masks (bottom). b) The flow of training our model, which generates a prediction of cell center points and biomarker expressions (Fake K) for end-to-end Ki67 scoring. c) Scoring information is extracted from generated Fake K, aggregated per slide, and compared with our diagnostic data.

## 2.2. IHCScoreGAN

### 2.2.1. PROXY TASK

Unsupervised nuclear cell segmentation and biomarker expression identification is a proxy task for our model, which builds off CycleGAN (Zhu et al., 2017). We train our model using a domain transfer strategy to learn how to convincingly transform our Ki67 stain input tiles $x$ into synthetic colored segmentation masks $y$.

The objective of our model is to learn an unpaired domain transfer between domains $\mathcal{X}$ and $\mathcal{Y}$, where we seek to train a generator $G : \mathcal{X} \rightarrow \mathcal{Y}$ which can transform an input $x \in \mathcal{X}$ following $\hat{y} = G(x)$, where $\hat{y} \in \hat{\mathcal{Y}}$. To enforce a robust mapping onto the distribution of $\mathcal{Y}$, an inverse generator $F : \mathcal{Y} \rightarrow \mathcal{X}$ is simultaneously trained to maintain "cycle consistency". Generator $F$ can transform an input $y \in \mathcal{Y}$ following $\hat{x} = F(y)$, where $\hat{x} \in \hat{\mathcal{X}}$. The idea is that an input $x$ should be penalized for being different from its reconstruction $\hat{x} = F(G(x))$ through a cycle consistency loss, and similarly for an input $y$ in the other mapping direction:

$$\begin{aligned} \mathcal{L}_{\text{CYC}_X} &= \mathbb{E}_x \left[ ||F(G(x)) - x||_1 \right] \\ \mathcal{L}_{\text{CYC}_Y} &= \mathbb{E}_y \left[ ||G(F(y)) - y||_1 \right] \end{aligned} \tag{1}$$

Next, the generated domain $\hat{\mathcal{Y}}$ is constrained to be indistinguishable from domain $\mathcal{Y}$, which is estimated by training discriminators $D_Y$ and $D_X$, which produce logits predicting

whether an input is from $\mathcal{Y}$ or $\hat{\mathcal{Y}}$, or from $\mathcal{X}$ or $\hat{\mathcal{X}}$, respectively, through an adversarial loss:

$$\mathcal{L}_{\mathrm{GAN}_X} = \mathbb{E}_x\left[\log(1 - D_Y(G(x)))\right] + \mathbb{E}_x\left[\log(D_X(x))\right]$$
$$\mathcal{L}_{\mathrm{GAN}_Y} = \mathbb{E}_y\left[\log(1 - D_X(F(y)))\right] + \mathbb{E}_y\left[\log(D_Y(y))\right] \quad (2)$$

where generators $G$ and $F$ are updated such that $\mathcal{L}_{\mathrm{GAN}_X}$ and $\mathcal{L}_{\mathrm{GAN}_Y}$ are minimized, and discriminators $D_X$ and $D_Y$ are updated such that $\mathcal{L}_{\mathrm{GAN}_X}$ and $\mathcal{L}_{\mathrm{GAN}_Y}$ are maximized.

Finally, an input $x$ should be identical to itself after mapping onto its own distribution $\hat{x} = F(x)$, and similarly for $y$, constrained through an identity loss:

$$\mathcal{L}_{\mathrm{IDT}_X} = \mathbb{E}_x\left[||F(x) - x||_1\right]$$
$$\mathcal{L}_{\mathrm{IDT}_Y} = \mathbb{E}_y\left[||G(y) - y||_1\right] \quad (3)$$

Our model's generators are each composed of encoder-decoder architectures similar to a four-block U-Net (Ronneberger et al., 2015). Our discriminators each resemble a contracting convolutional neural network. Network details are further elaborated in Appendix A.1.

### 2.2.2. Center Point and Cell Type Generation

We previously defined our target dataset as $S_{\mathrm{MASK}} = \{(y_i, k_i)\}_{i=1}^{N_{\mathrm{MASK}}}$, where $y$ is a synthetic colored segmentation mask tile and $k$ is its matching cell center point distance map and binary cell expression masks. In this section, we aim to predict $k \in \mathcal{K}$ from a given Ki67 input tile $x$, which contains the critical information for achieving simple end-to-end quantification.

To generate $\hat{k}$, we add a second decoder branch $\mathrm{DEC}_k$ to our generator $G$ so that it generates outputs $\{(\hat{y}, \hat{k})\} = G(x)$ through its main and secondary branches simultaneously. For clarity, we hereafter denote the output of $G$ through $\mathrm{DEC}_k$ as $\hat{k} = G_K(x)$, where $\hat{k} \in \hat{\mathcal{K}}$. Our generator thus resembles $G : \mathcal{X} \to \{(\mathcal{Y}, \mathcal{K})\}$. Our second decoder branch $\mathrm{DEC}_k$ follows the same structure and input as its main decoder branch $\mathrm{DEC}_y$, but we detach the input embeddings from the gradient graph in this branch, since the proxy task already embeds all necessary information for $\mathrm{DEC}_k$ in its encoder embeddings (i.e., cell expression and shape).

To train $\mathrm{DEC}_k$, we add three learning objectives to the model. First, we introduce an additional cycle consistency loss between real $k \in \mathcal{K}$ and reconstructed $\hat{k} \in \hat{\mathcal{K}}$:

$$\mathcal{L}_{\mathrm{CYC}_K} = \mathbb{E}_{y,k}\left[||G_K(F(y)) - k||_1\right] \quad (4)$$

To encourage generation of realistic cell center points and expressions, we add a new discriminator $D_K$ which produces logits predicting whether an input is from domain $\mathcal{K}$ or $\hat{\mathcal{K}}$, along with a new adversarial loss:

$$\mathcal{L}_{\mathrm{GAN}_K} = \mathbb{E}_x\left[\log(1 - D_K(G_K(x)]\right] + \mathbb{E}_k\left[\log(D_K(k))\right] \quad (5)$$

where generator $G_K$ is updated such that $\mathcal{L}_{\mathrm{GAN}_K}$ is minimized, and discriminator $D_K$ is updated such that $\mathcal{L}_{\mathrm{GAN}_K}$ is maximized. Discriminator $D_K$ follows the same internal structure as discriminators $D_X$ and $D_Y$.

Finally, we tie our identity constraint to the encoding of $y$ by evaluating $\hat{k} = G_K(y)$ and comparing it with its matching $k$:

$$\mathcal{L}_{\mathrm{IDT}_K} = \mathbb{E}_{y,k}\left[||G_K(y) - k||_1\right] \quad (6)$$

The model architecture and loss constraints introduced in this section encourage generation of convincing $\hat{k}$ through $\hat{k} = G_K(x)$, which is a simple, single-stage process.

### 2.3. Objective Function

Our model's overall objective function combines Equations 1 through 6:

$$\begin{aligned}
\mathcal{L}_{model} = {} & \lambda\mathcal{L}_{\mathrm{CYC}_X} + \beta\mathcal{L}_{\mathrm{GAN}_X} + \gamma\mathcal{L}_{\mathrm{IDT}_X} \\
& + \lambda\mathcal{L}_{\mathrm{CYC}_Y} + \beta\mathcal{L}_{\mathrm{GAN}_Y} + \gamma\mathcal{L}_{\mathrm{IDT}_Y} \\
& + \lambda\mathcal{L}_{\mathrm{CYC}_K} + \beta\mathcal{L}_{\mathrm{GAN}_K} + \gamma\mathcal{L}_{\mathrm{IDT}_K}
\end{aligned} \tag{7}$$

where $\lambda$, $\beta$, and $\gamma$ are scaling terms. We use $\lambda = 10$, $\beta = 1$, and $\gamma = 10$ in this paper, following CycleGAN (Zhu et al., 2017). $\mathcal{L}_{model}$ is simultaneously minimized with respect to generators $G$ and $F$ and maximized with respect to discriminators $D_X$, $D_Y$, and $D_K$.

### 2.4. End-To-End Scoring

During inference, we extract predicted cell center points and cell expressions from the generated $\hat{k} = G_K(x)$ using a local maxima algorithm (see Appendix A.2). We then aggregate the cell counts at the slide level and calculate the slide-level scoring following $\frac{\mathrm{Ki67}^+}{\mathrm{Ki67}^+ + \mathrm{Ki67}^-}$.

## 3. Results

### 3.1. Experiment Design

**Model Comparison** We compared our method against a state-of-the-art supervised model, DeepLIIF (Ghahremani et al., 2022), and a supervised U-Net (Ronneberger et al., 2015), both trained on the public **Breast Tumor Cell Dataset (BCData)** (Huang et al., 2020) using identical settings as their original paper. Dataset details are provided in Section 3.4.

**Error Metrics** Mayo Clinic pathologists used a 20% cut-off for diagnosing Ki67-high vs. Ki67-low cases based on the ASCO guideline (Giordano et al., 2022) in the clinical diagnosis reports. Therefore, in our internal data experiments, we used the same cut-off value to classify a case from each method's case-level Ki67 score prediction. We then compared results by evaluating precision, recall, accuracy, and F1 score. For our experiments on the public BCData dataset, we instead evaluated mean absolute error (MAE) of cell counts across images, used in the BCData paper. In the tables, arrows represent the direction of optimal performance for each metric. Error metric details are provided in Appendix A.3.

### 3.2. Internal Dataset Experiments

We performed a two-split experiment to reflect the performance for the model in a general use-case, where we randomly drew 1,532 cases in the train split and 594 cases in the test split, with tiles corresponding to the cases aggregated into their split. Our framework outperformed both supervised models on all error metrics except precision (Table 1).

We also evaluated the clinical use case of our proposed framework, mimicking deployment in a real-world setting where the model was trained on older data collected from 2012 to 2018 and then evaluated on cases collected after 2018. This division resulted in 1,532 cases in the train split and 594 cases in the test split, with tiles corresponding to the cases aggregated into their split. Each model exhibited a reduction in performance in this experiment – we observed that the after-2018 samples were more challenging due to an increase in background noise, such as darker stroma staining (Table 2 and Figure 2).

Table 1: Random train/test split; comparison against clinical diagnosis

| Model | Precision↑ | Recall↑ | Accuracy↑ | F1↑ |
|---|---|---|---|---|
| UNet | **1.00** | 0.67 | 0.92 | 0.80 |
| DeepLIIF | 0.95 | 0.71 | 0.92 | 0.81 |
| IHCScoreGAN | 0.94 | **0.96** | **0.97** | **0.95** |

Table 2: Chronological train/test split; comparison against clinical diagnosis

| Model | Precision↑ | Recall↑ | Accuracy↑ | F1↑ |
|---|---|---|---|---|
| UNet | **0.98** | 0.46 | 0.88 | 0.63 |
| DeepLIIF | 0.90 | 0.53 | 0.88 | 0.66 |
| IHCScoreGAN | 0.89 | **0.95** | **0.96** | **0.92** |

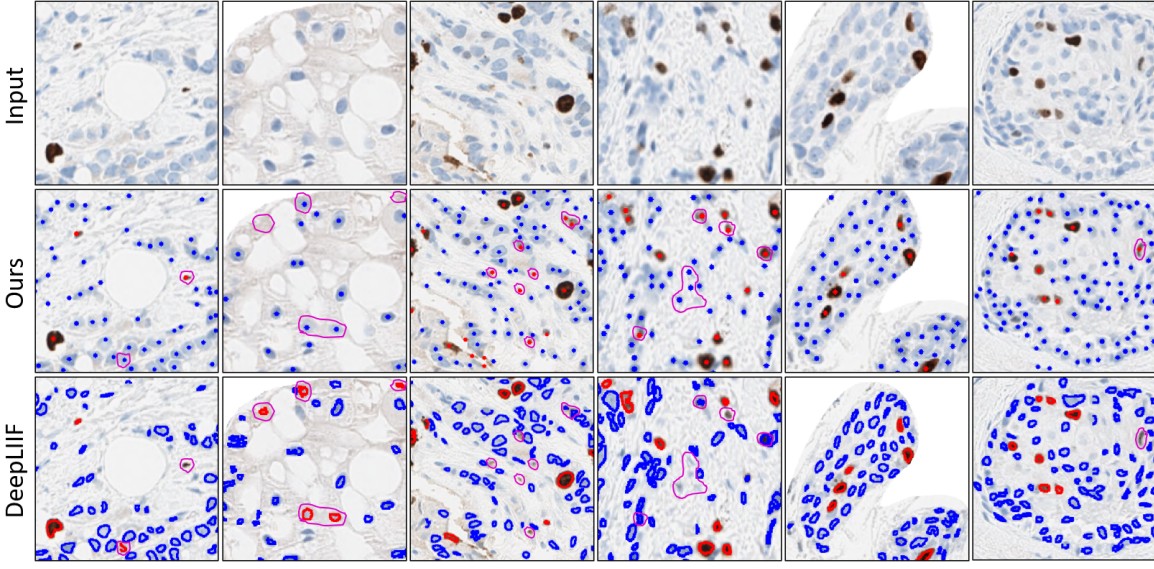

Figure 2: Qualitative comparison. Red and blue indicates positive and negative cell prediction, respectively; magenta indicates situations where the supervised model failed.

### 3.3. Cutoff Analysis

In practice, cases very close to the classification cutoff may be equivocal. Using the results in the clinical simulation experiment in Section 3.2, we relaxed the classification of equivocal cases by using each interval from ±1% to ±10% around the 20% ASCO cutoff point (Figure 3) and classifying cases both predicted and labeled within the interval as correct. Each model improved as the cutoff interval was relaxed, but we note that IHCScoreGAN exhibited near-optimal performance when relaxing the cutoff interval by just ±2%, reporting 0.93, 0.99, 0.98, and 0.96 in precision, recall, accuracy and F1 score, respectively (see Appendix B.1 for precise values).

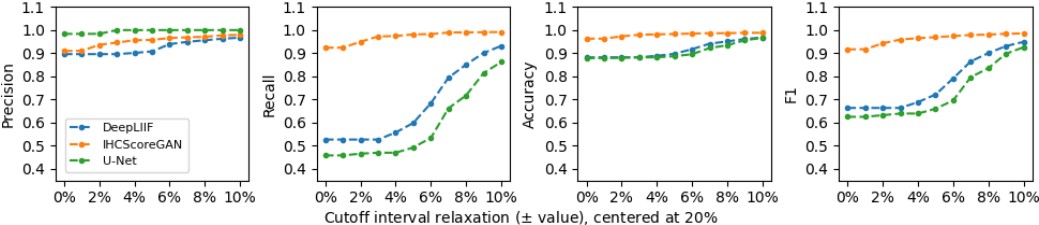

Figure 3: Effect of relaxing the case classification cutoff interval.

### 3.4. External Dataset Experiments

We validated our framework on a public breast cancer dataset, **BCData**, which features 1,338 Ki67 stain images taken at $40\times$ magnification by a Motic BA600-4 scanner, along with 181,074 manually annotated cell center points and cell types (Huang et al., 2020). BCData splits 803, 133, and 402 images for training, validation, and testing, respectively. We compared cell counting of our framework against two supervised models, both of which were trained using BCData's training images and cell annotation labels. The results are in Table 3, where 'MP', 'MN', and 'MA' represents the MAE of positive, negative, and the average value, respectively. Our unsupervised framework did not outperform the supervised state-of-the-art DeepLIIF, yet still achieved comparable performance without needing the annotation labels. Both ours and DeepLIIF outperformed all supervised baseline models reported in (Huang et al., 2020). It is possible that the training sample size limited our framework's performance, suggested by our sample size experiments in Appendix B.3.

Table 3: Experiments on the external BCData dataset; comparison of cell counting

| Model | Trained On | MP↓ | MN↓ | MA↓ |
|-------|-----------|------|-------|-------|
| UNet | BCData | 7.88 | 22.89 | 15.39 |
| DeepLIIF | BCData | **5.44** | **12.23** | **8.83** |
| IHCScoreGAN | Internal | 9.91 | 17.88 | 13.89 |
| IHCScoreGAN | BCData | 6.43 | 14.32 | 10.37 |

### 4. Discussion

In this work, we proposed the first unsupervised framework, IHCScoreGAN, for end-to-end Ki67 scoring. We validated our method on 2,126 breast cancer cases and showed high agreement with clinical diagnoses provided by our pathologists. Experimental comparison against pre-trained supervised methods on our internal dataset showed the significant advantage of our framework; pre-trained DeepLIIF often erroneously segmented stroma tissue and missed/misclassified challenging cells, likely resulting from differences from its training domain (Figure 2). On external data, we yielded close cell counting performance to the fully-supervised state-of-the-art DeepLIIF, and superior performance to other supervised models, without needing the training annotations. This work is limited by lacking comparisons with supervised models which are fine-tuned on our dataset, which is out of the scope of this work due to the complexity of fairly assessing such an experiment. We do not consider inter-observer variability in our clinical diagnoses in this work.

## Acknowledgments

We acknowledge all the support from the Division of Computational Pathology and AI, Department of Laboratory Medicine and Pathology, Mayo Clinic: Dr. Steven N. Hart, Debra A. Novak, Katelyn A. Reed, Dr. Daniel Macaulay.

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

## Appendix A. Additional Implementation Details

### A.1. Network Design

Our generators $G$ and $F$ follow encoder-decoder architectures similar to a four block U-Net (Ronneberger et al., 2015). The encoders consist of four contracting "down-convolution"

blocks, each consisting of a convolution layer (with a kernel size of 4, stride of 2, and padding of 1), followed by a LeakyReLU activation layer and an instance normalization layer. There are 64 convolution channels in the first encoder block, which increase by a factor of 2 in each block (up to 512 channels). The encoders have 8 additional residual blocks, each consisting of an in-place convolution (with kernel size 3), again followed by a LeakyReLU activation layer and an instance normalization layer. The decoders consist of 8 residual blocks of the same definition, followed by four expanding "up-convolution" blocks, each consisting of a transposed convolution layer (with a kernel size of 4, stride of 2, and padding of 1), followed by a LeakyReLU activation layer and an instance normalization layer. There are 512 convolution channels in the first decoder block, which decrease by a factor of 2 in the first three blocks (down to 64 channels) and end with a convolution onto 3 channels in the last decoder block. The outputs of each down-convolution block in the encoder is given a skip-connection into the corresponding up-convolution block in the decoder (i.e., the output of the third down-convolution block is concatenated with the output of the first up-convolution block, which is the input of the second up-convolution block). Finally, we perform a tanh activation on the outputs of the last decoder block, representing the final model prediction.

As mentioned in Section 2.2.2, generator $G$ additionally has a second decoder which follows identical structure to the main decoder described above, including identical skip-connections coming from the encoder, except that its gradients do not flow into the encoder during backpropagation.

Our discriminators $D_X$, $D_Y$, and $D_K$ consist of similar four-block encoders, described above, followed by a convolution layer (with a kernel size of 4, stride of 1, and padding of 1) onto a single output channel. The result is a patch-wise real/fake logit prediction.

## A.2. End-To-End Scoring Details

During inference, we achieve end-to-end scoring from the generated $\hat{k} = G_K(x)$ by using a simple local maxima algorithm on the cell center point maps for instance detection, followed by an $\arg\max$ operation for each detected instance across the binary type maps for cell expression prediction. We stitch together $\hat{k}$ before we run the extraction algorithm, which resolves prediction artifacts at the tile borders.

To formally define the local maxima algorithm, we further define $\{(\hat{m}, \hat{b})\} \in \hat{k}$, where $\hat{m}$ is the center point distance map and $\hat{b}$ is the group of binary type segmentation masks. The local maxima algorithm can then be defined as:

$$T = \{\arg\max(\hat{b}_{ij}) : (\hat{m}_{ij} > \omega) \wedge (\hat{m}_{ij} = \max_{i-\delta \leq h \leq i+\delta, j-\delta \leq w \leq j+\delta} \hat{m}_{hw})\} \tag{8}$$

where $\delta$ is a given neighborhood size, $\omega$ is a given threshold, and $(i, j)$ are indices of pixels within the center point distance map $\hat{m}$.

This is a common, simple 2D local maxima algorithm (e.g., (Brieu et al., 2019)). We used the `maximum_filter` function implemented in the `scipy` package in Python and assigned a pixel as a cell instance if it is its own local maxima (within $\delta$ pixels) and if it exceeds the threshold $\omega$. In our experiments, we used a neighborhood size $\delta$ of 25 and a threshold $\omega$ of 0.5.

### A.3. Formal Definition of Error Metrics

For our internal dataset experiments, we compare classification against a two-class ground truth, based on counting of cell types. For this binary classification problem, we evaluate precision, recall, accuracy, and F1 score, which are commonly used binary classification metrics. Let us define correctly-classified positive cases as True Positive (TP), correctly-classified negative cases as True Negative (TN), incorrectly-classified positive cases as False Negative (FN), and incorrectly-classified negative cases as False Positive (FP). We then formally define our binary classification error metrics:

$$
\begin{aligned}
\text{Precision} &= \frac{\text{TP}}{\text{TP} + \text{FP}} \\
\text{Recall} &= \frac{\text{TP}}{\text{TP} + \text{FN}} \\
\text{Accuracy} &= \frac{\text{TP} + \text{TN}}{\text{TP} + \text{TN} + \text{FP} + \text{FN}} \\
\text{F1 Score} &= \frac{2 * \text{TP}}{2 * \text{TP} + \text{FP} + \text{FN}}
\end{aligned}
\tag{9}
$$

For external dataset experiments on BCData, we used Mean Absolute Error (MAE) to compare cell counting performance. MAE was used by BCData authors to evaluate cell counting on their dataset (Huang et al., 2020), formally defined as:

$$
\text{MAE}^{category} = \frac{\sum_{i=1}^{n} |c_i^{category} - \hat{c}_i^{category}|}{n}
\tag{10}
$$

where *category* represents cell counts per biomarker expression or total cell counts on a per-image basis, and $c_i^{category}$ and $\hat{c}_i^{category}$ are the ground truth and predicted cell counts per category, respectively.

### A.4. Training Hyperparameters

In each experiment, we trained our model using 40,000 training iterations, a learning rate of 0.0002, a batch size of 4, and an Adam optimizer. Each value was selected through a multi-run grid search in order to optimize the model's quantitative performance on a separate internal test dataset which is distinct from the data used in the experiments in this work.

### A.5. Hardware Details

We trained our model and performed all experimental validations in this paper using an NVIDIA® RTX™ A4000 GPU with 16GB memory. Our CPU was a 12-core, 3.2 Ghz Intel® Xeon® w5-3435X.

## Appendix B. Supplementary Experimental Results

### B.1. Cutoff Analysis Details

This table corresponds to the exact values of the points in the plot in Figure 3.

Table 4: Effect of relaxing 20% Ki67-positive cutoff threshold for case stratification

| Cutoff | DeepLIIF | | | | IHCScoreGAN | | | |
|--------|-----------|--------|----------|------|-------------|--------|----------|------|
| | Precision | Recall | Accuracy | F1 | Precision | Recall | Accuracy | F1 |
| 20% | 0.39 | 0.86 | 0.67 | 0.53 | 0.89 | 0.95 | 0.96 | 0.92 |
| $20 \pm 1\%$ | 0.39 | 0.86 | 0.67 | 0.53 | 0.90 | 0.96 | 0.97 | 0.93 |
| $20 \pm 2\%$ | 0.40 | 0.88 | 0.68 | 0.55 | 0.93 | 0.99 | 0.98 | 0.96 |
| $20 \pm 3\%$ | 0.41 | 0.90 | 0.68 | 0.57 | 0.93 | 0.99 | 0.98 | 0.96 |
| $20 \pm 4\%$ | 0.43 | 0.93 | 0.69 | 0.59 | 0.94 | 0.99 | 0.98 | 0.97 |
| $20 \pm 5\%$ | 0.46 | 0.94 | 0.70 | 0.62 | 0.95 | 0.99 | 0.98 | 0.97 |
| $20 \pm 6\%$ | 0.48 | 0.95 | 0.70 | 0.64 | 0.96 | 0.99 | 0.99 | 0.98 |
| $20 \pm 7\%$ | 0.51 | 0.97 | 0.71 | 0.67 | 0.96 | 1.00 | 0.99 | 0.98 |
| $20 \pm 8\%$ | 0.53 | 0.98 | 0.72 | 0.69 | 0.97 | 1.00 | 0.99 | 0.98 |
| $20 \pm 9\%$ | 0.56 | 1.00 | 0.73 | 0.72 | 0.97 | 1.00 | 0.99 | 0.98 |
| $20 \pm 10\%$ | 0.60 | 1.00 | 0.75 | 0.75 | 0.98 | 1.00 | 0.99 | 0.99 |

## B.2. Ablation Studies

We evaluated the effectiveness of 1. including random color intensities when creating the synthetic color masks, and 2. detaching gradients from the inputs of the $\text{Dec}_K$ branch. We otherwise trained and evaluated our model in the same way as Section 3.2. We show overall better performance by including these elements of our proposed approach (Table 5).

Table 5: Model ablation studies; comparison against clinical diagnosis

| Model | Precision | Recall | Accuracy | F1 |
|-------|-----------|--------|----------|------|
| IHCScoreGAN, no color variation | **0.98** | 0.82 | 0.96 | 0.90 |
| IHCScoreGAN, no detach | 0.98 | 0.77 | 0.95 | 0.86 |
| IHCScoreGAN | 0.89 | **0.95** | **0.96** | **0.92** |

## B.3. Sample Size Experiments

We evaluated the data dependency of IHCScoreGAN by gradually increasing the number of randomly-selected samples seen by the model during training. We used the same train and test splits as in Section 3.2. To evaluate model stability, we trained our framework five separate times with the same training samples and then calculated the mean and standard deviation of each error metric. Our results are in Table 6, where $N$ represents the number of unique samples seen by the model during training.

## B.4. Cross Validation Studies

We evaluated a 5-fold cross validation of our model for case-level classification, where case-level folds are chosen randomly across our entire internal dataset. Specifically, our training splits consisted of approximately 1701 cases, and our testing splits consisted of approximately 425 cases, with the corresponding tiles aggregated into the respective split in each

Table 6: IHCScoreGAN sample size experiments; comparison against clinical diagnosis

| $N$ | Precision | Recall | Accuracy | F1 |
|---|---|---|---|---|
| 100 | 0.81±0.18 | 0.93±0.10 | 0.81±0.17 | 0.74±0.16 |
| 250 | 0.85±0.17 | 0.91±0.09 | 0.87±0.14 | 0.82±0.13 |
| 500 | 0.94±0.04 | 0.96±0.06 | 0.89±0.07 | 0.86±0.06 |
| 800 | 0.94±0.06 | 0.91±0.09 | 0.94±0.03 | 0.87±0.04 |
| 1000 | 0.91±0.05 | 0.94±0.08 | 0.90±0.06 | 0.89±0.03 |
| 2000 | 0.93±0.04 | 0.95±0.07 | 0.93±0.06 | 0.90±0.04 |
| 4000 | 0.89±0.07 | 0.92±0.08 | 0.93±0.05 | 0.89±0.02 |
| 8000 | 0.97±0.01 | 0.94±0.06 | 0.95±0.01 | 0.88±0.03 |
| 16000 | 0.96±0.05 | 0.95±0.06 | 0.93±0.03 | 0.88±0.03 |
| 32000 | 0.96±0.05 | 0.96±0.06 | 0.94±0.03 | 0.90±0.01 |
| 40000 | 0.95±0.06 | 0.95±0.06 | 0.94±0.03 | 0.89±0.03 |

K-fold experiment. We additionally evaluated DeepLIIF and U-Net on each testing split, for reference. The averaged results across all five splits is shown in Table 7.

Table 7: Cross-validation studies; comparison against clinical diagnosis

| Model | Precision | Recall | Accuracy | F1 |
|---|---|---|---|---|
| UNet | **0.99±0.01** | 0.63±0.03 | 0.91±0.01 | 0.77±0.02 |
| DeepLIIF | 0.96±0.02 | 0.67±0.04 | 0.92±0.01 | 0.79±0.02 |
| IHCScoreGAN | 0.94±0.09 | **0.90±0.06** | **0.96±0.02** | **0.91±0.03** |

