# OpenReview forum: "IHCScoreGAN: An unsupervised generative adversarial network for end-to-end ki67 scoring for clinical breast cancer diagnosis"
_MIDL.io/2024/Conference — MIDL 2024 Oral_

### Official Review · Reviewer_PGy6 · 2024-02-18

**Confidence:** 4
**Preliminary Rating:** 3
**Final Rating:** 4

**Summary:**

The authors propose a framework for label-free Ki67 assessment based on a GAN framework. This approach is claimed to significantly outperform a supervised state-of-the-art baseline (DeepLIIF). The authors provide additional experiments to further demonstrate the capabilities of their approach.

**Strengths:**

The major strength of the paper is its interesting method. By using HE-based pseudo-segmentation masks and unlabeled IHC images, the model learns to segment in the IHC. The model path that receives the unlabeled IHC images is likely to learn mainly the visual properties of the target area, thus mitigating any possible domain gap that would have to be overcome by a model trained on a different dataset and applied to the one at hand for inference.

**Weaknesses:**

The main weakness of the paper is the evaluation. Given the strong claims made by the authors, I would like to see a more comprehensive evaluation, in particular:
- The baseline against which the authors compare their algorithm is the DeepLIIF algorithm by Ghahremani et al. which is essentially an IHC segmentation model that, unlike the authors' method, has been trained in a supervised fashion. This model is applied to the test data, which is an in-house dataset of the authors. Therefore, no adaptation to the target domain could take place and the domain gap between training and target domain is likely to influence the results. However, the authors' method was trained on the target domain. For a fair comparison, either both models should be trained and tested on the same domain, or the authors could use a second domain as a hold out test set.
- To substantiate the authors' results, I would like to see an evaluation based on public datasets. The paper on the baseline approach refers to such a dataset, which also contains cell-level annotations (https://link.springer.com/chapter/10.1007/978-3-030-59722-1_28). Perhaps the authors could use this dataset to train a supervised baseline as well.
- Furthermore, I would like the authors to perform the training of their algorithm in a cross validation scheme to provide more information about the robustness of the algorithm. So far, only results on a training and test split have been provided. A cross-validation would dispel the doubt that the model performance depends on the distribution of the training and test data.

In the discussion, the authors could also elaborate on the results of Experiment 3.3. Why did the results of the baseline deteriorate even though it was not trained on the target domain?

Another weakness is the lack of details on the experimental design. How many pathologists were involved in the IHC scoring of the in-house dataset? Did they have to come to an agreement to establish a ground truth label? How were the 23 HE images for the target dataset selected from the TCGA?

**Detailed Comments:**

- As GANs are notoriously dependent on the choice of hyperparameters, please report numerical values for all the hyperparameters you used in the experiments.
- Please consider adding more baseline methods for your comparison, e.g., trained fully supervised.

**Justification Of Final Rating:**

The authors have clarified all major points that were raised by my review. I still think that the paper could profit from a more thorough investigation against more competitive approaches, and some minor doubts remain. But the paper in it's current is sufficient to be presented at MIDL.

**Justification Of The Preliminary Rating:**

The authors propose an interesting new method for IHC assessment by combining unlabeled IHC images and weak labels generated from HE images. However, given the strong claims made in the paper, I would like to see the authors' findings substantiated by further analysis.

**Questions To Address In The Rebuttal:**

- Please provide further evaluation on a second set of data.
- How does the authors' model perform on unseen data?
- If possible, please perform cross validation instead of single shot training.

---

> ### Author Response · Authors · 2024-03-18
> **Reviewer PGy6: Adding cross validation and external dataset experiments, adding baselines, and adding clarifications to the text.**
>
> *We would like to express our sincere acknowledgments and appreciation for reviewer’s efforts to review our paper. We value all the points made by the reviewer and addressed each of reviewer’s comments by adding multiple experiments, adding/editing content in the paper, providing further clarification in response.*
>
> ## RE: Weakness 1, 2; detailed comments 2; questions to address in rebuttal 1, 2.
>
> **The reviewer’s concerns are raised regarding lack of experimental comparisons by training and testing with sufficient baselines.**
>
> We addressed reviewer’s concern by:
> 1) Adding experiments (see section 3.4, results in Table 3) with two fully supervised baselines, DeepLIIF (state-of-the-art) and UNet (one of the most commonly used) on the public dataset (i.e., BCData dataset, suggested by the reviewer),
> 2) Adding results in Table 3 for our model trained in internal dataset and tested on the unseen data,
> 3) Re-performing validation using the BCData-trained supervised models (DeepLIIF and the added baseline, UNet) for comparison on our internal dataset (see updated results in Table 1 and 2).
>
> Our method demonstrated far superior performance on our internal-dataset at the case-level than the two supervised baselines. In the external dataset cell counting experiments, although IHCScoreGAN does not perform better than the state-of-the-art supervised model DeepLIIF, the results are comparable; additionally, both ours and DeepLIIF outperform all other supervised baseline models reported in the BCData paper. In Appendix B.3., we added additional sample size experiments which suggest that our method may not show its optimal performance considering the small patch-level sample size provided in the external dataset. In comparison, the supervised models were given more than 100,000 annotated cell instances, and DeepLIIF also required synthetic MxIF images in order to train.
>
> ## RE: Weakness 3, Question to address in the rebuttal 3.
>
> We addressed reviewer’s comments by adding five-fold cross-validation experiments; the results are shown in Appendix B.4 (to comply with the requirements for main body page limit of 8 pages). Considering that the experiments in Table 1 and 2 consist of different splits, and by adding the cross-validation experiments, we can confidently rule out the fact that our model performance is significantly dependent on the data distribution.
>
> ## RE: Detailed comments 1.
>
> As per the reviewer’s comment, we have included all choices of hyperparameters, as well as the selection method, in Appendix A.4.
>
> ## RE: "In the discussion, the authors could also elaborate on the results of Experiment 3.3. Why did the results of the baseline deteriorate even though it was not trained on the target domain?"
>
> **Please note that the previous Section 3.3 has been merged into Section 3.2 due to space constraints caused by the additional experiments.**
>
> We observed that the results deteriorate because the images after 2018 are more darkly stained, which is a more significant difference from the BCData dataset (which the supervised baselines trained on) than in images before 2018. We addressed reviewer’s comment by adding our observation in the text:
>
> 1) **Section 3.2:**  ‘...we observed that the after-2018 samples were more challenging due to an increase in background noise, such as darker stroma staining...’
>
> 2) **Section 4:**  ‘We observed that the pre-trained DeepLIIF often erroneously segmented
> stroma tissue and missed/misclassified challenging cells on our dataset, likely resulting from
> the differences in its training data (Figure 2).’
>
> ## RE:  "Another weakness is the lack of details on the experimental design. How many pathologists were involved in the IHC scoring of the in-house dataset? Did they have to come to an agreement to establish a ground truth label? How were the 23 HE images for the target dataset selected from the TCGA?"
>
> We added/edited the content to address reviewer’s comments:
> 1) **Section 2.1.1:**  “All cases in this study are sourced from Mayo Clinic and collected from 2012 through 2023 primarily by 6 pathologists. The Ki67 slides were scanned at 20× (0.5μm/pixel) resolution by Aperio® AT scanners. Clinical diagnoses were performed on selected invasive tumor regions. Typically, one pathologist is involved in the clinical diagnosis for each case.”
>
> 2) **Section 2.1.2:** “We selected 23 H&E tissue slides of 20× resolution such that a variety of cell sizes, structures, and shapes were represented, divided them into tiles of size 256 × 256, and sampled N_Mask = 10, 000 tiles from the tissue regions.”

---

> > ### Comment · Reviewer_PGy6 · 2024-03-19
> > **Response to the review**
> >
> > Dear authors,
> >
> > Thank you for the detailed responses to my comments. However, I still have some questions regarding the submission.
> >
> > - Regarding the selection of hyperparameters in Appendix A.4, you state that “ Each value was selected through a multi-run grid search in order to optimize the model’s quantitative performance on our internal testing data.” Can you please clarify what you mean by internal testing data? Did you ensure that this data did not overlap with your actual test data split to prevent your model/hyperparameters from overfitting the test data?
> > - Why do the authors still report the results of the two-split experiment (3.2) while they already have done a five-fold cross validation (B.4)? The results of the cross validation have a much higher statistical informativeness as the results are far less likely to be influenced by the data distribution of the training and test splits. The authors could also report the mean scores of the supervised baselines on the test sets of the five fold cross validation.
> > - Regarding the selection of the 23 images from the TCGA, I see that you have added some information. However, I'm still concerned that the section (2.1.2) is missing crucial information. Since the TCGA is a large data repository with many datasets from different types of tumors, which datasets did you use to select the images? Did the images have a consistent visual appearance or did they appear to come from multiple laboratories? Why did the authors use only 23 images instead of an entire dataset? What was the motivation for using exactly 23 images?

---

> > > ### Author Response · Authors · 2024-03-25
> > >
> > > *We again sincerely thank the reviewer for their useful and valuable comments.*
> > >
> > > ## RE: Question 1
> > >
> > > We appreciate the reviewer’s comment and agree that this information is missing in the current version.
> > >
> > > Our internal testing data refers to a separate dataset that is independent of the dataset we used in our experiments. Therefore, our internal testing data does not overlap with any data used in our experiments.
> > >
> > > We propose to clarify this point in the camera-ready version: **“Each value was selected through a multi-run grid search in order to optimize the model’s quantitative performance on a separate internal test dataset which is distinct from the data used in the experiments in this work.”**
> > >
> > > ## RE: Question 2
> > >
> > > We sincerely appreciate reviewer’s input on our paper. We would like to further discuss our thoughts as below:
> > >
> > > We value the reviewer’s suggestion, therefore we added the 5-fold cross-validation results to our paper. Regarding the two-split experiments, we believe that it has very important for potential clinical applications: The first (i.e., random two-split) is to be used as a performance baseline while the second (chronological two-split) reflects the data drift issue that is common in a real clinical setting (this is based on the guidance from our expert pathologist). We want to show how our model and other pre-trained models perform in such a situation.
> > >
> > > We believe that both the two-split and cross-validation experiments are very valuable, so we included both. Per the reviewer’s further suggestion in this round, we are working on generating results for the cross-validation for the supervised baselines and will include the results in the camera-ready version of our paper upon acceptance.
> > >
> > > ## RE: Question 3
> > >
> > > We appreciate the reviewer's comments and we would like to clarify as below:
> > >
> > > We randomly selected those WSIs from the Breast Invasive Carcinoma (TCGA-BRCA) project’s data. The 23 were manually chosen to include a variety of cell sizes, structures, and shapes. Visually, we found those images varied to a certain extent, but we cannot confirm if they are from different laboratories based on the information from TCGA, and we did not select images with consideration to image variability or laboratory source. We qualitatively assessed the quality of the generated target dataset. TCGA-BRCA consists of 1,098 cases, each of which may consist of multiple tissue slides. Considering the training efficiency and usability of our method, we try to use a small sample size for generating the target dataset.
> > >
> > > To address the reviewer’s concerns, enhance our method’s reproducibility, and explain the selection process in more detail, we suggest providing target masks along with the released code, and additionally propose an update to the camera-ready version upon paper acceptance to the following:
> > >
> > > **“We randomly sampled H\&E tissue slides of $20\times$ magnification from the Breast Invasive Carcinoma project (TCGA-BRCA) and manually selected slides to represent a variety of cell sizes, structures, and shapes (i.e., to discourage the model from using these features to discriminate real vs. fake data), resulting in 23 slides.”**

---

> > > > ### Comment · Reviewer_PGy6 · 2024-03-26
> > > > **Response to the comments**
> > > >
> > > > I value the authors' responsiveness to the criticisms. I acknowledge that the paper has significantly improved during the review process and I will incorporate this assessment into my final evaluation.

---

### Official Review · Reviewer_5rTw · 2024-02-28

**Confidence:** 3
**Preliminary Rating:** 4
**Recommendation:** Poster
**Final Rating:** 4

**Summary:**

Deep learning models that have been trained in a fully supervised fashion for Ki67 quantification rely on hard-to-get annotations and may be prone to overfitting on training data distribution. The authors propose a method to overcome these issues : using unlabeled IHC image samples and unpaired synthetic data, they train a Cycle-GAN-inspired network to precict cell expression. Their method outperforms a fully supervised baseline trained on another dataset and tested in a generalization setup, that is on their in-house data.

**Strengths:**

- The paper is overall well-written, structured and complete. The experiments included are relevant and enable the proposed method to be properly evaluated, qualitatively and quantitatively.

- The presence of appendices is very useful for reproducibility and for adding details to experiments.

- The expriment on the clinical robustness of the model is much appreciated.

**Weaknesses:**

- The method is only evaluated on a private dataset, which restricts the reproducibility of the experiments.

- The proposed model is only compare to a single supervised other model from the litterature, in a generalization setup. It does not include comparison with fine tuning on few samples or comparison with other adaptation techniques.

- The authors insist that the model does not require annotations, and is thus adapted to overcome the limit of having distinct training and testing data distrubtions. However, it still required a collection of over 1,500 images to be trained, which could limit the use of the model to centers that have not acquired as much data.

**Detailed Comments:**

- I can't quite intuite why detaching the Decoder K from the gradients map help improving the performance of the model. Do you have an idea of why such phenomenon appears ?

- In the discussion paragraph, you claim that the unsupervised model, because it is less prompt to overfit on the training data distribution, learn more general rules and representation of the data. A experiement on how your model generalize to unseen data domains (not onyl the shift in data collection year) would help validating this hypothesis.

- At inference do you use a stride that is equal to the tile size, or do you have an overlap ?

**Justification Of Final Rating:**

I gave the paper a rating of 4, which I will leave as it, with more confidence than before. I would like to emphasize that the authors put a great effort into answering the question that were addressed to them. They have added experiments and details that have strenghtened the quality of their work.

**Justification Of The Preliminary Rating:**

The paper is very clear and of good quality. The proposed method gives very good results and helps overcoming the need of fine annotations to train or fine-tune fully supervised model that may only be suited for particular in-house data distribution. However, it would benefit from further experimentation to compare it with baseline methods.

**Questions To Address In The Rebuttal:**

- How were the values of the $\lambda$, $\beta$ and $\gamma$ loss scaling parameters determined ? Were they taken from a previous or determined by a grid search ? The information does not appear in the main text or in the appendices, although it seems important to me, given the number of terms in the cost function and the known instability of GANs training.

- You compare your method with a pre-trained fully supervised baseline, but not with any other domain adaptation methods. Other paradigms of training could be used to overcome the issue that you're tackling of obtaining annotations (few-shot/unsupervised/self-supervised adaptation, domain alignment, weak adaptations, image-level labels, etc.).

- You evaluate the performance of your model using two-split validation. Have you tried evaluating the robustness and uncertaincy of your model to multiple training ?

**Special Issue:**

No

---

> ### Author Response · Authors · 2024-03-18
> **Reviewer 5rTw: Adding sample size, multi-run and external dataset experiments, adding baselines, and addressing questions.**
>
> ## RE: Weakness 1, 2 and Question address in rebuttal 2
>
> To provide fair comparison with more baselines and improve reproducibility, we addressed the reviewer’s concerns by:
> 1) Adding/updating our experiments, and
> 2) Providing intention for publishing our code.
>
> We performed experiments on a public dataset instead of performing domain adaptation on our internal dataset since this provides better transparency and reproducibility alongside the publication of our code.
>
> We added an experiment using a public external breast cancer dataset, BCData, where we trained and evaluated our framework against two fully supervised baselines, DeepLIIF and U-Net, for cell counting (see Section 3.4, results in Table 3). We also added/updated experiments on our internal dataset: we updated the results in Tables 1 and 2 for DeepLIIF, now trained on BCData, for consistency across all of our experiments. We also added another supervised baseline, U-Net, also trained on BCData, to Tables 1 and 2 in order to provide more baselines to address reviewer’s concerns. Our method demonstrated far superior performance on our internal-dataset at the case-level than the two supervised baselines. In the external dataset cell counting experiments, although IHCScoreGAN does not perform better than the state-of-the-art supervised model DeepLIIF, the results are comparable; additionally, both ours and DeepLIIF outperform all baseline supervised models reported in the BCData paper. In Appendix B.3., we added additional sample size experiments which suggest that our method may not show its optimal performance considering the small patch-level sample size provided in the external dataset.
>
> We also plan to publish our code, and added a placeholder link to the abstract:
>
> 'Upon publication, our code will be made available at www.github.com.'
>
> This should significantly strengthen our reproducibility. Given the unsupervised nature of our model, any researcher with access to our codebase can effortlessly train our model on their own dataset, which offers another layer of transparency.
>
> ## RE: Question address in rebuttal 1.
>
> We addressed this comment by:
> 1) Adding the information in the paper, and
> 2) Providing intention for publishing our code (see above).
>
> We added the following information to the main text:
> 1) **Section 2.3.:** "We use $\lambda=10$, $\beta=1$, and $\gamma=10$ in this paper, following CycleGAN."
> 2) **Appendix A.4.:** "Each value was selected through a multi-run grid search in order to optimize the model's quantitative performance on our internal testing data."
>
> ## RE: weakness 3, Question address in rebuttal 3:
>
> We addressed reviewer’s comments by:
> 1) Providing clarification, and
> 2) Adding additional sample size experiments (see appendix B.3 and results in Table 6, appendix B.4).
>
> We regret that we were not clear about the data splitting. Our chronological data splits happened to consist of 1,532 cases before 2018, and we wanted to make the two-split experiment comparable, thus we split the data similarly. With added sample size experiment in appendix section B.3, Table 6, we provided a multi-run sample size experiment to demonstrate that we only need 1,000-2,000 images to achieve strong and relatively stable performance. In addition, the added 5-fold cross validation results in section B.4  also reflect the evaluation of the robustness and uncertainty of our model. We observed small standard deviation in both experiments in general.
>
> ## Re: detailed comments1:
>
> The two branches constitute different tasks: the proxy task branch wants to construct a mask so that the original image can be reconstructed, while the secondary branch wants to find cell center points and cell expressions. If we do not detach the gradients of Decoder K, the reconstruction task becomes much more difficult due to competing goals; e.g., the proxy task branch wants to encode cell intensity to aid reconstruction, while the secondary branch does not want this information since it is orthogonal to cell expression prediction.
>
> ## Re: detailed comments2:
>
> We performed new experiments in Table 3, where our model trained with internal data and test on the unseen public external data, as the reviewer suggested. Our results show our model performs moderately. To avoid confusion, we removed our speculation in the paper.
>
> ## Re: detailed comments3:
>
> In our framework, we do not use an overlap, but instead keep the stride equal to the tile size; instead, we stitch together cell center point maps and cell expression masks before running our center point extraction algorithm, since nearby predictions from adjacent tiles are resolved by the local maxima algorithm.
>
> We added the following information to the paper to clarify:
> 1) **Appendix A.2.:** "We stitch together ˆk according to their spatial position before we run the extraction algorithm, which resolves prediction artifacts at the tile borders."

---

> > ### Comment · Reviewer_5rTw · 2024-03-26
> > **Response**
> >
> > I would like to thank the authors for having taken into account my recommandations and addressed my concerns with carefulness and quality.

---

### Official Review · Reviewer_kX4u · 2024-03-02

**Confidence:** 3
**Preliminary Rating:** 3
**Recommendation:** Poster
**Final Rating:** 4

**Summary:**

This paper aims to quantify percentage of ki67-positive cells in IHC slide by training a CycleGAN to convert samples between IHC tiles to/from synthetic data generated from H&E slides and cell segmentation labels. The authors demonstrated their approach is better than one unsupervised baseline in Ki67 scoring.

**Strengths:**

+ The paper is pretty easy to read. The figures are super helpful in understanding what is going on.
+ This paper demonstrated strong performance when compared to unsupevised baselines, i.e., DeepLIIF.

**Weaknesses:**

+ The paper over-claims that its approach can adapt to real-world imaging variabilities. The authors should add experiments demonstrating the approach works well on a wide range of IHC datasets, e.g., collected from other institutions, with different scanners, varying tissue qualities and staining protocols. If these experiments could not be done, the authors should avoid claiming that this is a main contribution of the paper.
+ Another concern is the lack of comparisons to supervised learning baseline, or any baseline that is finetuned on the in-house dataset that is used to train IHCScoreGAN. The authors should perhaps try to do such comparison on open source datasets where labels are available and provide additional baselines that at least is finetuned on the in-house data for fair comparison to the proposed approach.

**Detailed Comments:**

n/a

**Justification Of Final Rating:**

- The authors clarified that the main contribution is not a robust method that adapts to datasets from different centers, but instead an unsupervised solution that does not rely on annotations for good performance. This is a totally valid point that I misunderstood.
- The authors also added experiments that demonstrate the proposed approach is worse but comparable to DeepLIIF on an external datasets where annotations are available. This shows promise.
- The authors also point out the paper's clinical utility which I think is super important that is not in the paper but good to know such context.  I would think it's hard to know whether the proposed method is useful for other people, I would still want to encourage this kind of research where the project is motivated by a clear clinical need.

Given the above points, I raise the rating to 4.

**Justification Of The Preliminary Rating:**

This paper solves a real clinical problem and achieves pretty good performance in ki67-scoring! However, the authors somewhat overclaims the approach's robustness & lacks important baselines. I would recommend borderline.

**Questions To Address In The Rebuttal:**

Try to address points made in weaknesses.

**Special Issue:**

No

---

> ### Author Response · Authors · 2024-03-18
> **Reviewer kX4u: Adding baselines, external dataset experiments, and claim clarifications.**
>
> *We would like to express our sincere acknowledgments and appreciation for reviewer’s efforts to review our paper. We value all the points made by the reviewer and addressed each of reviewer’s comments.*
>
> ## RE: Weakness No.1:
>
> To address reviewer’s concern, we:
> 1) Clarified our statement, and
> 2) Added experiments that validates our model on an external public dataset.
>
> **Clarification**
>
> This is the specific statement we wrote originally:
>
> *"By removing ground truth requirements, our unsupervised technique constitutes an important step towards easily-trained, generalizable Ki67 scoring solutions that can adapt to real-world imaging variability and be further adapted to other cancer types or protein biomarkers."*
>
> We did not claim that our method has already achieved such achievement, but instead said 'important step towards'. We intended to say that our method provided a new unsupervised approach without needing labels (i.e., domain adaptation in this case). To make this point clearer, we edited our statement in the paper to address reviewer’s concern regarding the overclaim.
>
> *"By removing ground truth requirements, our unsupervised technique constitutes an important step towards easily-trained, generalizable Ki67 scoring solutions which can train on out of domain data in an unsupervised manner."*
>
> Although we insist that our original statement was not an overclaim, we agree with the reviewer that additional experiments would add value to our work. As such, we added substantial experiments to address reviewer’s concern on this point (see section 3.4; results in Table 3) using a public external dataset, for which all procedures for obtaining the image are drastically different from our internal dataset. Our model shows comparable performance with the state-of-the-art fully supervised method and substantially better performance than U-Net. In Appendix B.3., we added additional sample size experiments, which suggest that our method may not show its optimal performance considering the small patch-level sample size provided in the external dataset. In comparison, the supervised models were trained on more than 100,000 cell instances, and DeepLIIF also required synthetic MxIF images in order to train. We hope that these experiments aptly address the concerns raised by the reviewer.
>
> ## RE: Weakness No.2:
>
> We addressed reviewer’s concern by following reviewer’s advice/suggestions, which by adding experiments (Section 3.4 and Table 3) on a public (i.e., open source) external dataset with two fully supervised models/baselines: DeepLIIF, which is the state-of-the-art on the BCData dataset; and U-Net, which is one of the most commonly used and rigorously studied models for cell segmentation for biomedical images.
>
> We also added/edited experiments for internal dataset, we updated the results in Table 1, 2 and 3 for DeepLIIF with model trained on the breast cancer dataset (BCData). We also added U-Net results, also trained on BCData, in Tables 1-3 to provide more baselines to address reviewer’s concern. Our method demonstrated far superior performance on our internal-dataset at case-level that the two supervised baselines; it also showed comparable performance to the fully supervised state-of-the-art and substantially better performance than the U-Net baseline on the open source dataset at cell-counting level; although, again, our method may not show its optimal performance considering its unsupervised methodology.
>
> Importantly, the focus of our paper is the clinical oriented task, so we focused on a single clinical test with the intention for integration for our clinical workflow (see paper title and content of our paper). We have no intention to showcase/prove that our model is the best model/a better model than other models after all. Instead, our experiments intend to show the gap between a pre-trained model for potential clinical use vs. our unsupervised approach. Based on the instruction from MIDL website, MIDL appreciates clinical use cases, and our paper links a strongly-performing method with substantial novelty to a clinical use case validation, thus the validation should be sound  on its own considering its excellent performance on a substantially large clinical dataset (also supported by the comment from reviewer No.2's comments on the strengths). In this case, our work should not be punished by its validation. However, to make sure we address reviewer’s concern, we still added substantial experiments (for both internal and external datasets) with strong performance.
>
> We also plan to publish our code; we added a placeholder link to the abstract:
>
> 'Upon publication, our code will be made available at www.github.com.'
>
> Given the unsupervised nature of our model, any researcher with access to our codebase can effortlessly train our model on their own dataset, which offers another layer of transparency.

---

### Meta-Review · Area_Chair_imAq · 2024-04-03

**Recommendation:** Accept (Oral)
**Confidence:** 5

**Metareview:**

This paper received the final ratings with 3 weak accept. The authors addressed most of the comments during the rebuttal. Given the consensus of reviewers, AC recommends to accept the paper and strongly suggest the authors take the reviewers’ comments into the final revision.

---

### Decision · Program_Chairs · 2024-04-05

Accept (Oral)